# An interpretable data augmentation framework for improving generative modeling of synthetic clinical trial data

**Afrah Shafquat** [1]   **Jason Mezey** [1 2 3]   **Mandis Beigi** [1]   **Jimeng Sun** [4 1]   **Chufan Gao** [4 1]   **Jacob W. Aptekar** [1]

## Abstract

Synthetic clinical trial data are increasingly being seen as a viable option for research applications when primary data are unavailable. A challenge when applying generative modeling approaches for this purpose is many clinical trial datasets have small sample sizes. In this paper, we present an interpretable data augmentation framework for improving generative models used to produce synthetic clinical trial data. We apply this framework to three clinical trial datasets spanning different disease indications and evaluate the impact of factors such as initial dataset size, generative algorithm, and augmentation scale on metrics used to assess synthetic clinical trial data quality, including fidelity, utility, and privacy. The results indicate that this framework can considerably improve the quality of synthetic data produced using generative algorithms when considering factors of high interest to end users of synthetic clinical trial data.

## 1. Introduction

Synthetic clinical trial data provides an innovative solution for sharing clinical trial data(Z. Azizi & Collaborators, 2021)(H.-H. Wong & Eyraud, 2014), where patient privacy(Kulynych, 2008) and trial sponsor proprietary information concerns (Shafquat et al.) are major deterrents to sharing primary data. Given these constraints are the norm for clinical trial data, there is an increasing demand for synthetic data for research applications including designing more efficient clinical trials through revising dosage,

anticipating adverse events, and other strategies that can result in higher probabilities of a successful trial (Z. Azizi & Collaborators, 2021)(H.-H. Wong & Eyraud, 2014). To generate synthetic data for related clinical data sources such as Electronic Health Records (EHR), Generative Adversarial Networks (GAN) and Tabular Variational Autoencoders (TVAE)(Xu et al., 2019) have been successfully applied (Ghadeer Ghosheh & Zhu, 2022)(Muhammad Sakib Khan Inan & Uddin, 2023)(Giannis Nikolentzos & Brandt, 2023). However, most generative models used to generate synthetic clinical data have been evaluated and benchmarked on datasets where the training dataset size is large, often including tens to hundreds of thousands of patient records. A challenge for applying these same approaches to clinical trials is the datasets may only include only a few hundreds to thousands of patients depending on the disease indication and phase of the clinical trial(NIH, 2023). A consequence is the quality of generative modeling synthetic clinical trial data can be insufficient when considering data metrics of interest to consumers of these data, including data fidelity (Emam et al., 2021)(Beigi et al., 2022), utility (e.g. survival curves(Altman, 1992)(MK Goel & Kishore, 2010)), and privacy (Beigi et al., 2022)(Shafquat et al.).

For limitations on training data in other fields, such as computer vision and language models, data augmentation has been proposed as a potential solution (Gunjan Ansari & Saxena, 2021)(Steven Y. Feng & Hovy, 2021)(Tomoki Ishikawa & Urushihara, 2022)(Rasool Fakoor, 2020)(Hung & Gan, 2021)(Khoshgoftaar, 2019). Rudimentary techniques in computer vision (flipping, rotation, and cropping) and natural language models (oversampling with synonym substitution) have been used to create augmented datasets, where these approaches can improve performance for machine-learning based prediction models (Gunjan Ansari & Saxena, 2021)(Steven Y. Feng & Hovy, 2021)(Tomoki Ishikawa & Urushihara, 2022)(Rasool Fakoor, 2020)(Hung & Gan, 2021)(Khoshgoftaar, 2019). Similarly, augmentation-like techniques for tabular data have been explored using methods such as oversampling at random (Engelmann & Lessmann, 2020) and synthetic minority oversampling (SMOTE) (Nitesh V. Chawla & Kegelmeyer, 2002) to address issues of underrepresentation and bias for minority classes. However, challenges have been encountered with these ap-

---

[*]Equal contribution  [1]Medidata Solutions, a Dassault Systèmes company, New York, NY 10014 [2]Department of Computational Biology, Cornell University, Ithaca, NY 14850 [3]Department of Genetic Medicine, Weill Cornell Medicine, New York, NY 10065 [4]Computer Science Department, University of Illinois Urbana-Champaign Champaign, IL 61820. Correspondence to: Afrah Shafquat <Afrah.Shafquat@3ds.com>.

proaches including over-training on the augmented data (A. S. Tarawneh & Almuhaimeed, 2022) and bias when introducing duplicates in the data (Katherine Lee & Carlini, 2022)(Yuki Eizuka & Suzuki, 2021). While methods for correcting these issues have been explored including application-specific changes to the learning architecture and data augmentation approaches, the extension of these strategies to generative modeling of clinical trial data is not straightforward where methods like SMOTE (which upsample the "minority" classes in the data) or oversampling may introduce privacy concerns (A. S. Tarawneh & Almuhaimeed, 2022)(Katherine Lee & Carlini, 2022)(Yuki Eizuka & Suzuki, 2021)(Shafquat et al.).

In this paper, we propose an interpretable data augmentation framework to generate high-quality synthetic clinical trial data using generative models trained on limited data. Our framework takes advantage of the fact that the characteristics of the original dataset important to the end user for downstream data analysis, including easy to interpret metrics such as fidelity, utility, and privacy, can be directly controlled in the augmented dataset. The framework consists of two main phases: (i) augmentation of the clinical trial data and (ii) generation of synthetic data using the augmented training data. For (i), it is critical to use an algorithm that is able to generate an augmentation dataset that preserves interpretable metrics such as fidelity, utility, and privacy where "bad augmentations" can lead to significant decreases in the overall synthetic data quality (Chi-Heng Lin & Muthukumar, 2022). For generating augmentation datasets for clinical trial data, we use "Simulants" (Beigi et al., 2022) a dimension reduction, re-sampling method designed to produce synthetic data interpretably similar to the original, by preserving utility, fidelity and privacy. We evaluate the improvement in the performance of generative synthetic algorithms when trained on Simulants augmented clinical trial datasets. Though any generative data algorithm can be used in this framework, in the current set of experiments, we evaluate this data augmentation framework on methods that have previously been used to generate other types of clinical data with large sample sizes (i) `CTGAN`, (ii) `CopulaGAN`, and (iii) `TVAE` (Patki et al., 2016). To the best of our knowledge, a data augmentation framework to train generative models to generate synthetic data has not been explored to date. This augmentation framework presents a first time exploration of this concept.

## 2. Conditions for augmentation

When using a GAN or VAE to generate synthetic data, the overall goal is to train a latent variable generator or decoder of a distribution $Q$ that is suitably close to $P$, the unknown true distribution of the input data $\mathbf{X}_N$ of sample size $N$.

The problem is when $P$ is complex and the $N$ of the input data is relatively small, the generative model $Q$ may not be close enough to $P$, where the output may not be useful for a given application. Of interest is deciding under what conditions training on augmented data $\mathbf{X}'_{aN}$ from an unknown distribution $P' \neq P$ with samples size $a * N$, produced by some mechanism, will result in a better outcome than when training on $\mathbf{X}_N$. Considering the generator $G(Z)$ of a GAN with latent variables $Z$ as the example, and utilizing Kullback-Leibler divergence $D_{KL}(P||Q)$ to assess the difference between distributions, the condition when augmented data will result in an improved outcome can be phrased as:

$$\mathbb{E}_{G(Z)'|\mathbf{X}'_{aN} \sim P'} D_{KL}(P||Q') < \mathbb{E}_{G(Z)|\mathbf{X}_N \sim P} D_{KL}(P||Q) \tag{1}$$

where $G(Z)$ is the generator resulting from input data $\mathbf{X}_N$, $G(Z)'$ is the generator of $Q'$ resulting from input data $\mathbf{X}'_{aN}$, and the expectations are over possible data augmented data sets sampled from $P'$ and possible observed input data sampled from $P$. For this expression, in the most extreme limiting case where $a \to \infty$ and assuming training of networks in the GAN with arbitrary width and / or depth (Hanin & Sellke, 2017), this produces:

$$D_{KL}(P||P') < \mathbb{E}_{G(Z)|\mathbf{X}_n \sim P} D_{KL}(P||Q) \tag{2}$$

illustrating that in the limit, an augmented data approach has value only if $P$ and $P'$ are suitably close. In many applications, producing augmented data where $D_{KL}(P||P')$ is small enough can be challenging, where we might expect this distance to increase with increasing $a$. In such cases, as increased training occurs on the augmented samples, the resulting output may not be useful.

An advantage when considering the objective of producing synthetic clinical trial data is the suitability of data for the end user often depends on quantifiable metrics such as fidelity, utility, and privacy. Considering a fidelity metric $\mathbb{F}$ (Emam et al., 2021)(Beigi et al., 2022)(Patki et al., 2016) as the example, where larger values indicate greater similarity of two datasets or distributions, use of augmented data will result in higher fidelity synthetic data when:

$$\mathrm{E}_{G(Z)'|\mathbf{X}'_{aN} \sim P'} \mathbb{F}(P,Q') > \mathbb{E}_{G(Z)|\mathbf{X}_N \sim P} \mathbb{F}(P,Q) \tag{3}$$

As $\mathbb{E}_{G(Z)'|\mathbf{X}'_{aN} \sim P'} \mathbb{F}(P,Q')$ is equal to $\mathbb{E}_{G(Z)'|\mathbf{X}'_{aN} \sim P'} \mathbb{F}(P',Q')$ when $P' = P$ and, for any $P'$, is an increasing function of $\mathbb{E}_{G(Z)'|\mathbf{X}'_{aN} \sim P'} \mathbb{F}(P',Q')$ and is equal to $\mathbb{F}(P,P')$ as $a \to \infty$, if we assume strong control of $\mathbb{F}(P,P')$ to high values, then

$$\mathrm{F}(\mathrm{P, P'}) + \mathrm{c} \ \mathrm{E}_{G(Z)'|\mathbf{X}'_{aN} \sim P'} \mathbb{F}(P',Q') > \mathbb{E}_{G(Z)'|\mathbf{X}'_{aN} \sim P'} \mathbb{F}(P,Q') \tag{4}$$

where the constant $c$ is determined by $\mathbb{F}(P,P')$ and will tend to be small when $\mathbb{F}(P,P')$ is strongly controlled. Use

of augmented data can therefore improve the fidelity of synthetic data when:

$$\mathbb{F}(P, P') + c \; \mathbb{E}_{G(Z)'|\mathbf{X}'_{aN} \sim P'}\mathbb{F}(P', Q') > \mathbb{E}_{G(Z)|\mathbf{X}_N \sim P}\mathbb{F}(P, Q) \tag{5}$$

Note that if this condition is met, in cases where $\mathbb{F}(P, P')$ is bounded relatively close to the maximum, as $a$ increases, then fidelity of the synthetic data will increase. Conversely, if $\mathbb{F}(P, P')$ varies with $a$, an intermediate value of $a$ may provide the best result. In sum, as the objective when using generative models to produce synthetic clinical trial data that has high fidelity (or related) to the original data, and since the fidelity of the augmented and original data can be quantified, if strongly controlling observed fidelity leads to strong control of fidelity on the underlying distributions, augmented data can improve synthetic data quality.

## 3. Experiments

### 3.1. Experimental Setup

The data augmentation framework (Figure 1) involves (i) generation for a high-quality augmented training dataset (as compared to the original data) through an augmentation algorithm, (ii) training a generative model (e.g. `CTGAN`) using the augmented training dataset, and (iii) generating synthetic patient records using the trained generative model. The improvement through this augmentation framework is evaluated by synthesizing clinical trial data generated using only the original clinical trial data as input versus using augmented training data at different scales. This framework is evaluated across three clinical trial datasets: (i) Multiple Myeloma, (ii) Non-small cell lung cancer, and (iii) CAR-T, while using generative synthetic data algorithms: (i) `CTGAN`, (ii) `CopulaGAN`, and (iii) `TVAE`.

### 3.2. Data augmentation framework

The framework of data augmentation involves the following steps:

1. For each clinical trial dataset $\mathbf{X}_N$ containing $N$ samples, the Simulants data augmentation algorithm generates an augmented dataset $\mathbf{X}'_{aN}$, where $a$ the "augmentation scaling factor" can be varied such that the size of the resulting augmented dataset is $a * N$. For the current set of experiments, augmentation scaling factor $a$ is set to 2, 3, 6 or 11.

2. For each original clinical trial dataset $\mathbf{X}_N$ and augmented dataset $\mathbf{X}'_{aN}$, we train a generative model $\mathbf{G}$ to generate synthetic data $S(\mathbf{X}_N)$ and $S(\mathbf{X}'_{aN})$ where all synthetic datasets contain $N$ samples (to allow fair comparison across datasets).

3. Compare synthetic data performance metrics (fidelity,

*Table 1.* Description of clinical trial datasets

| DATASET | NO. PATIENTS | NO. FEATURES |
|---|---|---|
| MULTIPLE MYELOMA | 946 | 14 |
| NON-SMALL CELL LUNG CANCER | 3001 | 24 |
| CAR-T | 5619 | 19 |

utility and privacy) on $S(\mathbf{X}_N)$ vs. $\mathbf{X}_N$ and $S(\mathbf{X}'_{aN})$ vs. $\mathbf{X}'_{aN}$ where improved values in metrics for $S(\mathbf{X}'_{aN})$ indicate improvement in synthetic data quality due to augmentation.

For increased reliability of results, multiple synthetic datasets ($n = 3$) are generated for each clinical trial dataset, generative algorithm, and augmentation scaling factor combination. Performance metrics (utility, fidelity and privacy evaluation) and standard deviation are reported in Section 4.

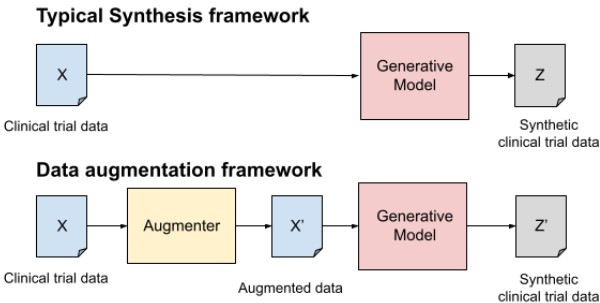

*Figure 1.* Data augmentation framework for synthetic data generation

### 3.3. Data sources

The impact of data augmentation is evaluated on three clinical trial datasets that are referred to as: (i) Multiple Myeloma, (ii) Non-small cell lung cancer, and (iii) CAR-T. Each clinical trial dataset is an aggregation of multiple trials for the specific disease indication denoted by the dataset's indicated name (except for CAR-T). The CAR-T dataset consists of patients who participated in clinical trials where the Chimeric Antigen Receptor T-cell therapy (CAR-T) was administered. Table 1 lists the number of patients and features for each clinical trial dataset.

### 3.4. Synthetic data generation algorithms

We use the Synthetic Data Gym (SDGym) from the Synthetic Data Vault (SDV) Project (Patki et al.) (Patki et al., 2016) to evaluate and compare the effect of our data augmentation method on the data generated by three state-of-

the-art synthesizers, `CopulaGAN`, `CTGAN`, and `TVAE` (Tabular Variational AutoEncoder) (Patki et al., 2016)(Xu et al., 2019). The SDV benchmark is a library which offers a set of classical and novel synthetic data generators to use as comparative baselines as well as a large collection of evaluation metrics for cross-validation of the synthetic data against the original data. Generative Adversarial Networks (GANs) are deep learning based generative models which consist of two independent sub-models: the generator and the discriminator (i.e. adversary). As the generator generates new data, the discriminator distinguishes real data from the generated data. `CTGAN` is a GAN-based architecture that models tabular data distributions. It addresses the challenges with using GANs on tabular data such as mixed data types consisting of highly imbalanced categorical data as well as data having non-gaussian distribution. `CopulaGAN` is a variation of `CTGAN` which utilizes Cumulative Distribution Function (CDF)-based transformation to facilitate the `CTGAN` model training. Variational Autoencoder (VAE) is another neural network based generative model which consists of an encoder and a decoder where the encoder maps the data to a latent space and the decoder performs the opposite to transform the data from the latent space to the input space. `TVAE` is an adapted VAE for tabular data by using the mixed-type data preprocessing and modifying the loss function. In the current set of experiments, the Synthetic Data Vault Python framework (version 1.0.0) is used to generate synthetic datasets for clinical trial datasets.

### 3.5. Augmentation algorithm

The algorithm Simulants is used to create augmentation datasets by generating synthetic clinical trial data that are interpretably similar to the augmented data in terms of fidelity, while preserving the privacy of the original data(Beigi et al., 2022). The "Simulants" algorithm employs the following steps:

1. Assume the original source data contains $n$ records, where each record contains $m$ distinct features regarding a subject, with features that can be a mix of categorical and numerical variables.

2. The original source data is encoded using one-hot-encoding and the missing values are imputed using any well-known imputation method.

3. Next the pairwise correlation coefficient is calculated between all pairs of features to determine the highly correlated features, where these features will be co-segregated when generating synthetic data.

4. The data is then embedded into a low-dimensional (e.g., 2 or 3 dimensions) feature space using PCA (Principal Component Analysis) (F.R.S., 1901) or other embedding approach (e.g., t-SNE (t-Stochastic Neighbor

Embedding)(der Maaten L & G., 2008), UMAP (Uniform Manifold Approximation and Projection) (Leland McInnes & Melville)).

5. After embedding the data, the $k$ nearest neighbor algorithm(Cover & Hart, 1967)(Buitinck et al., 2013) is applied, and for each point one (or more) synthetic data points are simulated by randomly permuting the features of its nearby neighbors within a certain radius/distance.

6. A multiplicative Gaussian error with a truncated distribution to the range of the features is added, where this error is centered on the feature value of each simulated record. For discrete features, the values are rounded to the closest integer values.

### 3.6. Fidelity evaluation tests

Fidelity metrics provide an interpretable quantity for evaluating the degree to which the synthetic data resemble the original data. The following fidelity metrics are used in this study: (i) bag of words distance (Beigi et al., 2022), (ii) Column Shapes (Patki et al., 2016), and (iii) Paired correlation $r^2$ (Patki et al., 2016).

- **Bag of words distance**: This metric measures the Euclidean distance between the centroid frequencies in real vs. synthetic datasets. Centroids are defined by K-Means clustering (Lloyd, 1982) on the real dataset. Centroid frequency in a dataset (real or synthetic) is defined as the frequency of samples that are closest to each centroid (as defined by the real dataset). Specifically, the bag of words distance measures the Euclidean distance between the centroid frequencies as computed for the real dataset compared to the synthetic data, where scores close to 0 are ideal.

- **Column shapes**. This metric measures the mean column similarity between real vs. synthetic data. Here the similarity between numerical columns is defined using the Kolmogrov-Smirinov test (Massey Jr, 1951) where the similarity is defined as a 1-KS-test statistic. The similarity for categorical columns is defined as the sum of the absolute differences in frequencies of values within the column between synthetic data and real data. The overall "Column Shapes" metric is calculated as the mean similarity across columns for the synthetic data. Values close to 1 indicate high similarity between the columns in synthetic compared to real data. This metric is sourced from Python SDV library (Patki et al., 2016).

- **Paired correlations** $r^2$. This metric indicates the coefficient of determination $r^2$ between pairwise correlations across columns between real vs. synthetic data.

This metric uses Pearson correlation and total variation distance for numerical and categorical columns respectively (Freedman et al., 2007). Values close to 1 indicate high similarity in pairwise correlations in synthetic compared to real data. This metric computes $r^2$ on paired correlations across features in the dataset sourced from the Python Synthetic Data Vault library (Patki et al., 2016).

### 3.7. Privacy evaluation tests

The following intuitive privacy metrics are used to calculate the risk presented by the synthetic datasets created using the augmented training datasets vs. the original training datasets.

- **Singling out**. This test assesses the risk of singling out a record using a unique univariate value for a specific record. Lower values close to 0 indicate a lower risk of disclosure using this attack. Python library `anonymeter` (Giomi et al., 2023) is used to evaluate this metric.

- **Linkability risk**. This test assesses the risk of linking a record from the real data to the synthetic data using a set of auxiliary information available such as age, sex, and race. Lower values close to 0 indicate a lower risk of disclosure using this attack. Python library `anonymeter` (Giomi et al., 2023) is used to evaluate this metric.

- **Mean inference risk**. This test assesses the mean inference risk across all columns in the synthetic data where inference risk for each column is computed as the attacker's ability to predict the attribute for an original record using the synthetic data. Lower values close to 0 indicate a lower risk of disclosure using this attack. Python library `anonymeter` (Giomi et al., 2023) is used to evaluate this metric.

- **Privacy loss**. This metric is calculated as the difference between the adversarial test accuracy and adversarial train accuracy (Yale et al., 2019). Here, accuracy is defined as the attacker's ability (accuracy) in predicting if a record belongs to (i.e. is closest to) the real data compared to synthetic data where "Adversarial train accuracy" is computed on training real data (as real data) and "Adversarial test accuracy" on test real data. Here, higher accuracy indicates the attacker is able to differentiate between synthetic and real data easily. For privacy loss, values closer to 0 denote low privacy loss whereas high values close to 1 indicate higher loss of privacy.

### 3.8. Utility evaluation tests

The utility metrics for clinical trial datasets focus on performance of survival analysis and computation of survival probabilities and their differences between the original and synthetic datasets. Larger differences in survival probabilities indicate poor utility and smaller differences indicate higher utility resulting in the end-user inferring similar conclusions when they use the original data compared to the synthetic data.

- **Log-rank test**: The logrank test(Bland & Altman, 2004) is used to test the null hypothesis that there is no difference between the populations in the probability of an event (i.e. death) at any time point. The analysis is based on the times of events (i.e. death). P-value of the log-rank test indicates the probability that the test-statistic was computed by chance. In this case, higher p-values are better and indicate higher similarity between the original and synthetic dataset. Lower p-values indicate low probability that the differences between the survival curves were observed by chance, which is not ideal.

- **Survival probabilities** $r^2$ : Coefficient of determination $r^2$ is computed between the survival probabilities at fixed time intervals in the original dataset compared to the synthetic data. $r^2$ values ranged from 0 to 1 where high $r^2$ values close to 1 indicated higher utility compared to lower $r^2$ values. High $r^2$ values also indicate strong alignment between the respective survival probabilities (original vs. synthetic).

## 4. Results and Discussion

### 4.1. Fidelity evaluation results

Fidelity of synthetic data to original data was defined using the following fidelity metrics: (i) bag of words distance (Beigi et al., 2022), (ii) Column shapes (Patki et al., 2016), and (iii) Paired correlation $r^2$ (Patki et al., 2016). The fidelity metrics are computed on synthetic data generated using the original training data denoted as "Original" or augmented training data denoted as "Augmented 2x", "Augmented 3x", "Augmented 6x", or "Augmented 11x" where the number indicates the "augmentation scaling factor" as described in Section 3.2. Tables 2, 3, and 4 show the mean fidelity metrics across the three synthetic datasets produced for each clinical trial dataset (Multiple Myeloma, Non-small cell lung cancer, and CAR-T) while using each generative synthetic data algorithm (`CTGAN`, `CopulaGAN`, and `TVAE`). Fidelity metrics for the data produced using the algorithm Simulants (which is used for augmentation) are reported for reference in each of the fidelity plots. The results across fidelity metrics indicate that augmented training

*Table 2.* Fidelity evaluation for Multiple Myeloma. Best score across models per algorithm is in **bold**. Best overall score is underlined.

| MODEL (INPUT) | DISTANCE | COLUMN SHAPES | CORRELATION $r^2$ |
|---|---|---|---|
| CTGAN (ORIGINAL) | 0.29 (0.01) | 0.83 (0.02) | -0.14 (0.07) |
| CTGAN (AUGMENTED 2X) | 0.28 (0.04) | 0.83 (0.01) | 0.03 (0.10) |
| CTGAN (AUGMENTED 3X) | 0.27 (0.01) | 0.86 (0.01) | 0.07 (0.04) |
| CTGAN (AUGMENTED 6X) | 0.18 (0.00) | **0.89 (0.01)** | 0.58 (0.05) |
| CTGAN (AUGMENTED 11X) | **0.10 (0.02)** | 0.88 (0.00) | **0.79 (0.00)** |
| COPULAGAN (ORIGINAL) | 0.33 (0.03) | 0.75 (0.01) | -0.03 (0.08) |
| COPULAGAN (AUGMENTED 2X) | 0.30 (0.01) | 0.72 (0.03) | 0.07 (0.15) |
| COPULAGAN (AUGMENTED 3X) | 0.28 (0.03) | 0.79 (0.01) | 0.23 (0.16) |
| COPULAGAN (AUGMENTED 6X) | 0.19 (0.02) | 0.80 (0.01) | 0.65 (0.03) |
| COPULAGAN (AUGMENTED 11X) | **0.10 (0.01)** | **0.81 (0.01)** | **0.77 (0.03)** |
| TVAE (ORIGINAL) | 0.11 (0.01) | 0.84 (0.00) | 0.90 (0.04) |
| TVAE (AUGMENTED 2X) | 0.08 (0.00) | 0.85 (0.01) | 0.92 (0.01) |
| TVAE (AUGMENTED 3X) | 0.08 (0.01) | **0.86 (0.00)** | **0.94 (0.01)** |
| TVAE (AUGMENTED 6X) | **0.07 (0.01)** | 0.84 (0.02) | 0.93 (0.02) |
| TVAE (AUGMENTED 11X) | **0.07 (0.02)** | 0.85 (0.00) | 0.89 (0.06) |
| SIMULANTS (ORIGINAL) | 0.01 | 0.97 | 0.92 |

*Table 3.* Fidelity evaluation for Non-small cell lung cancer. Best score across models per algorithm is in **bold**. Best overall score is underlined.

| MODEL (INPUT) | DISTANCE | COLUMN SHAPES | CORRELATION $r^2$ |
|---|---|---|---|
| CTGAN (ORIGINAL) | 0.27 (0.02) | 0.91 (0.01) | 0.06 (0.11) |
| CTGAN (AUGMENTED 2X) | 0.23 (0.01) | 0.91 (0.02) | 0.39 (0.02) |
| CTGAN (AUGMENTED 3X) | 0.23 (0.01) | 0.92 (0.01) | 0.66 (0.02) |
| CTGAN (AUGMENTED 6X) | 0.17 (0.00) | **0.94 (0.01)** | 0.81 (0.02) |
| CTGAN (AUGMENTED 11X) | 0.09 (0.00) | 0.93 (0.00) | **0.88 (0.01)** |
| COPULAGAN (ORIGINAL) | 0.29 (0.04) | 0.87 (0.02) | 0.24 (0.11) |
| COPULAGAN (AUGMENTED 2X) | 0.24 (0.00) | 0.88 (0.00) | 0.44 (0.05) |
| COPULAGAN (AUGMENTED 3X) | 0.22 (0.01) | 0.88 (0.01) | 0.67 (0.03) |
| COPULAGAN (AUGMENTED 6X) | **0.15 (0.00)** | **0.93 (0.00)** | 0.80 (0.02) |
| COPULAGAN (AUGMENTED 11X) | 0.19 (0.01) | 0.90 (0.01) | **0.88 (0.01)** |
| TVAE (ORIGINAL) | 0.17 (0.01) | 0.83 (0.00) | 0.83 (0.06) |
| TVAE (AUGMENTED 2X) | 0.17 (0.01) | **0.89 (0.00)** | 0.90 (0.00) |
| TVAE (AUGMENTED 3X) | 0.16 (0.01) | **0.89 (0.01)** | 0.95 (0.01) |
| TVAE (AUGMENTED 6X) | 0.10 (0.01) | **0.89 (0.01)** | **0.98 (0.01)** |
| TVAE (AUGMENTED 11X) | **0.07 (0.03)** | 0.88 (0.00) | 0.97 (0.01) |
| SIMULANTS (ORIGINAL) | 0.00 | 0.99 | 0.98 |

*Table 4.* Fidelity evaluation for CAR-T. Best score across models per algorithm is in **bold**. Best overall score is underlined.

| MODEL (INPUT) | DISTANCE | COLUMN SHAPES | CORRELATION $r^2$ |
|---|---|---|---|
| CTGAN (ORIGINAL) | 0.19 (0.02) | 0.86 (0.00) | 0.61 (0.02) |
| CTGAN (AUGMENTED 2X) | 0.15 (0.01) | 0.87 (0.01) | 0.68 (0.04) |
| CTGAN (AUGMENTED 3X) | 0.12 (0.01) | **0.89 (0.03)** | 0.84 (0.00) |
| CTGAN (AUGMENTED 6X) | 0.11 (0.00) | 0.88 (0.02) | 0.83 (0.04) |
| CTGAN (AUGMENTED 11X) | **0.09 (0.00)** | 0.88 (0.00) | **0.86 (0.02)** |
| COPULAGAN (ORIGINAL) | 0.21 (0.02) | 0.77 (0.01) | 0.65 (0.02) |
| COPULAGAN (AUGMENTED 2X) | 0.15 (0.00) | 0.77 (0.01) | 0.70 (0.06) |
| COPULAGAN (AUGMENTED 3X) | 0.13 (0.01) | **0.84 (0.00)** | **0.86 (0.02)** |
| COPULAGAN (AUGMENTED 6X) | **0.09 (0.00)** | 0.78 (0.00) | 0.82 (0.03) |
| COPULAGAN (AUGMENTED 11X) | **0.09 (0.00)** | 0.82 (0.00) | **0.86 (0.03)** |
| TVAE (ORIGINAL) | 0.10 (0.01) | 0.85 (0.01) | 0.89 (0.03) |
| TVAE (AUGMENTED 2X) | 0.13 (0.01) | 0.85 (0.01) | 0.87 (0.02) |
| TVAE (AUGMENTED 3X) | 0.13 (0.01) | **0.86 (0.01)** | 0.86 (0.01) |
| TVAE (AUGMENTED 6X) | 0.17 (0.01) | 0.85 (0.00) | **0.90 (0.02)** |
| TVAE (AUGMENTED 11X) | **0.07 (0.01)** | **0.86 (0.00)** | 0.89 (0.01) |
| SIMULANTS (ORIGINAL) | 0.01 | 0.99 | 1.00 |

Non-small cell lung cancer and the GAN-based models compared to TVAE and CAR-T. The modest improvement in CAR-T may be explained by the initial dataset size of over 5000 samples, suggesting that the improvement due to training data augmentation may be of most importance where initial dataset size is smaller than 5000. Further investigation is needed to define the upper limits in improvement relative to scale of the initial dataset available. Across increasing values of augmentation scaling factor from 2 to 11, increased improvements across most of the fidelity metrics (Bag of words distance, Paired correlation $r^2$) and clinical trial datasets are also observed.

### 4.2. Utility evaluation results

Utility of synthetic clinical trial data is evaluated using the Kaplan-Meier curves (Altman, 1992)(MK Goel & Kishore, 2010) fitted on the time to death and death flag variables present in each clinical trial dataset. Survival analysis is performed on the original dataset and the synthetic datasets created by each of the synthetic data algorithms trained on augmented training data vs. original clinical trial data only.

Tables 5, 6, and 7 present the results of the log-rank test for synthetic data produced using augmented training data compared to original clinical trial data. As shown for most of the datasets, synthetic data generated using augmented data had lower log-rank test statistic and $p > 0.05$ (where statistical significance $\alpha = 0.05$). All synthetic data generated using the original data resulted in higher log-rank test statistics and $p < 0.05$ (except for CTGAN and CopulaGAN for Multiple Myeloma, and only CopulaGAN for CAR-T). None of the synthetic datasets generated for Non-small lung cancer using the original clinical trial data only resulted in log-rank test $p > 0.05$ across the synthetic data algorithms. In contrast, most synthetic datasets where augmented training datasets were used result in log-rank test $p > 0.05$ across clinical trial datasets and synthetic data algorithms.

For additional evidence, the $r^2$ for survival probabilities

data improved the fidelity of the synthetic data produced over those produced by training only on the original clinical trial dataset.

The overall fidelity of synthetic data depends highly on the synthesizer algorithm used, where TVAE outperformed other algorithms (CTGAN, CopulaGAN) across clinical trial datasets when using only original data as the training input. When trained with augmented training datasets, however, the quality of the synthetic data improved across synthetic data algorithms indicating the power of data augmentation for GAN models. This can be observed for CTGAN where the "Paired Correlation $r^2$" is less than 0.1 for synthetic data generated using the original clinical trial data as input (mean $r^2 = -0.14$ for Multiple Myeloma; mean $r^2 = 0.06$ for Non-small lung cancer). In contrast, the "Paired Correlation $r^2$" increases to around 0.8 for synthetic data generated using CTGAN with augmented data as input (mean $r^2 = 0.79$ for Multiple Myeloma; mean $r^2 = 0.88$ for Non-small cell lung cancer). This trend was consistent across fidelity metrics and clinical trial datasets where the largest improvements are seen in Multiple Myeloma and

*Table 5.* Utility evaluation for the Multiple Myeloma dataset. Best score across models per algorithm is in **bold**. Best overall score is underlined.

| MODEL (INPUT) | TEST STATISTIC | $p$ | $r^2$ |
|---|---|---|---|
| CTGAN (ORIGINAL) | 13.23 (18.52) | 0.31 (0.52) | 0.92 (0.08) |
| CTGAN (AUGMENTED 2X) | 21.63 (29.02) | 0.02 (0.02) | 0.89 (0.12) |
| CTGAN (AUGMENTED 3X) | 21.84 (8.85) | 0.00 (0.00) | 0.88 (0.06) |
| CTGAN (AUGMENTED 6X) | **1.93 (1.72)** | **0.32 (0.38)** | **0.96 (0.02)** |
| CTGAN (AUGMENTED 11X) | 13.21 (9.45) | 0.04 (0.06) | 0.95 (0.05) |
| COPULAGAN (ORIGINAL) | 60.45 (89.93) | 0.06 (0.11) | 0.75 (0.34) |
| COPULAGAN (AUGMENTED 2X) | 46.71 (65.68) | 0.06 (0.10) | 0.76 (0.27) |
| COPULAGAN (AUGMENTED 3X) | 70.55 (33.98) | 0.00 (0.00) | 0.71 (0.13) |
| COPULAGAN (AUGMENTED 6X) | **6.76 (10.42)** | **0.28 (0.27)** | **0.94 (0.02)** |
| COPULAGAN (AUGMENTED 11X) | 13.81 (7.39) | 0.00 (0.00) | 0.94 (0.02) |
| TVAE (ORIGINAL) | 21.89 (11.39) | 0.00 (0.00) | 0.94 (0.03) |
| TVAE (AUGMENTED 2X) | 7.89 (4.03) | 0.01 (0.01) | 0.97 (0.01) |
| TVAE (AUGMENTED 3X) | **3.27 (2.39)** | **0.12 (0.10)** | **0.99 (0.00)** |
| TVAE (AUGMENTED 6X) | 11.99 (6.36) | 0.00 (0.00) | 0.97 (0.02) |
| TVAE (AUGMENTED 11X) | 8.57 (2.42) | 0.01 (0.01) | 0.97 (0.01) |
| SIMULANTS (ORIGINAL) | 0.01 | 0.94 | 1.00 |

*Table 6.* Utility evaluation for the Non-small cell lung cancer dataset. Best score across models per algorithm is in **bold**. Best overall score is underlined.

| MODEL (INPUT) | TEST STATISTIC | $p$ | $r^2$ |
|---|---|---|---|
| CTGAN (ORIGINAL) | 97.87 (118.94) | 0.00 (0.00) | 0.83 (0.14) |
| CTGAN (AUGMENTED 2X) | 47.87 (42.17) | 0.05 (0.08) | 0.90 (0.06) |
| CTGAN (AUGMENTED 3X) | 21.08 (12.72) | 0.00 (0.01) | 0.96 (0.01) |
| CTGAN (AUGMENTED 6X) | **8.52 (12.17)** | **0.30 (0.45)** | **0.97 (0.02)** |
| CTGAN (AUGMENTED 11X) | 13.74 (3.69) | 0.00 (0.00) | 0.97 (0.01) |
| COPULAGAN (ORIGINAL) | 142.54 (128.43) | 0.00 (0.00) | 0.73 (0.20) |
| COPULAGAN (AUGMENTED 2X) | 21.18 (16.32) | 0.02 (0.03) | 0.94 (0.04) |
| COPULAGAN (AUGMENTED 3X) | **1.52 (1.39)** | **0.40 (0.45)** | 0.97 (0.01) |
| COPULAGAN (AUGMENTED 6X) | 5.22 (1.45) | 0.03 (0.02) | **0.98 (0.00)** |
| COPULAGAN (AUGMENTED 11X) | 8.73 (7.34) | 0.08 (0.13) | 0.98 (0.01) |
| TVAE (ORIGINAL) | 222.24 (88.57) | 0.00 (0.00) | 0.69 (0.11) |
| TVAE (AUGMENTED 2X) | **4.33 (2.26)** | 0.06 (0.07) | **0.99 (0.00)** |
| TVAE (AUGMENTED 3X) | 8.04 (10.72) | **0.32 (0.52)** | 0.98 (0.01) |
| TVAE (AUGMENTED 6X) | 6.48 (4.50) | 0.07 (0.11) | **0.99 (0.00)** |
| TVAE (AUGMENTED 11X) | 5.22 (5.73) | 0.24 (0.37) | 0.98 (0.00) |
| SIMULANTS (ORIGINAL) | 0.02 | 0.88 | 1.00 |

computed at fixed time intervals was computed for the original data and the synthetic data. As shown in Tables 5, 6, and 7, the $r^2$ values for synthetic data generated using augmented training data was higher than the $r^2$ values for synthetic data generated using only the original clinical trial data across synthetic data algorithms and clinical trial datasets. The largest increase in $r^2$ was observed in CTGAN where the mean $r^2 = 0.17$ (Table 7) for synthetic data generated using the original data compared to mean $r^2 = 0.87$ for the synthetic data generated using the augmented data (Model: "Augmented 2x"). Though improvements in fidelity using augmented data may be modest for CAR-T, the increase in the utility scores highlights the importance of using high-quality augmented data. Figure 2 shows an example of the Kaplan-Meier curves from the experiment results (Non-small lung cancer; "Augmented 5x"). As shown, synthetic datasets (red) generated using the augmented data produced survival curves with higher overlap with the original dataset (blue). In contrast, the curves generated using the original training data only deviated significantly from the original trend in the survival curves.

### 4.3. Privacy evaluation results

Risk of privacy disclosure presented by the synthetic data was evaluated using (i) Singling out risk(Giomi et al., 2023), (ii) Linkability risk(Giomi et al., 2023), (iii) Mean inference risk(Giomi et al., 2023), and (iv) Privacy Loss(Yale et al., 2019). The risk for each of the privacy metrics is computed for synthetic data created using the original clinical trial data compared to augmented clinical trial data as input to the synthesizer. Across most privacy metrics, the risk presented by synthetic data from generative models trained on augmented data was higher than that presented by the original data, though the absolute risk across metrics remained low.

Though the absolute risk presented by these synthetic data

*Table 7.* Utility evaluation for the CAR-T dataset. Best score across models per algorithm is in **bold**. Best overall score is underlined.

| MODEL | TEST STATISTIC | $p$ | $r^2$ |
|---|---|---|---|
| CTGAN (ORIGINAL) | 18.70 (6.52) | 0.00 (0.00) | 0.17 (0.40) |
| CTGAN (AUGMENTED 2X) | **5.43 (4.78)** | **0.20 (0.33)** | **0.87 (0.07)** |
| CTGAN (AUGMENTED 3X) | 35.76 (51.26) | 0.09 (0.16) | 0.58 (0.51) |
| CTGAN (AUGMENTED 6X) | 70.60 (45.61) | 0.00 (0.00) | 0.42 (0.35) |
| CTGAN (AUGMENTED 11X) | 45.38 (14.52) | 0.00 (0.00) | 0.78 (0.16) |
| COPULAGAN (ORIGINAL) | 16.83 (17.61) | 0.09 (0.15) | 0.47 (0.44) |
| COPULAGAN (AUGMENTED 2X) | 15.62 (13.16) | 0.02 (0.03) | 0.79 (0.16) |
| COPULAGAN (AUGMENTED 3X) | **6.97 (8.85)** | **0.13 (0.14)** | 0.81 (0.06) |
| COPULAGAN (AUGMENTED 6X) | 9.98 (6.35) | 0.03 (0.05) | **0.91 (0.07)** |
| COPULAGAN (AUGMENTED 11X) | 50.83 (23.86) | 0.00 (0.00) | 0.70 (0.25) |
| TVAE (ORIGINAL) | 44.95 (9.41) | 0.00 (0.00) | 0.47 (0.07) |
| TVAE (AUGMENTED 2X) | 26.45 (13.73) | 0.00 (0.00) | 0.62 (0.18) |
| TVAE (AUGMENTED 3X) | 44.53 (13.40) | 0.00 (0.00) | 0.60 (0.12) |
| TVAE (AUGMENTED 6X) | 17.45 (11.31) | 0.00 (0.01) | **0.76 (0.02)** |
| TVAE (AUGMENTED 11X) | **6.97 (6.92)** | **0.12 (0.19)** | 0.70 (0.09) |
| SIMULANTS (ORIGINAL) | 0.66 | 0.42 | 0.99 |

across privacy metrics remains low (less than 0.1), use of generative algorithms that incorporate differential privacy (e.g. DP-GAN(Liyang Xie & Zhou, 2018)) can provide an additional layer of privacy when using the proposed data augmentation framework. When considering privacy loss(Yale et al., 2019), TVAE has the largest privacy loss (less than 2%) indicating that overall the adversarial attack is not able to distinguish between a train and test dataset having similar success of attack in each scenario.

## 5. Conclusion

In this paper, we present a data augmentation framework that significantly improves the performance of generative models when generating synthetic clinical trial data. The overall performance of the framework highly depends on the synthetic data generation algorithm of choice; hence, generative models need to be carefully tested. We chose Simulants as the augmentation method as it can upsample the data to large amounts while still maintaining very high fidelity so

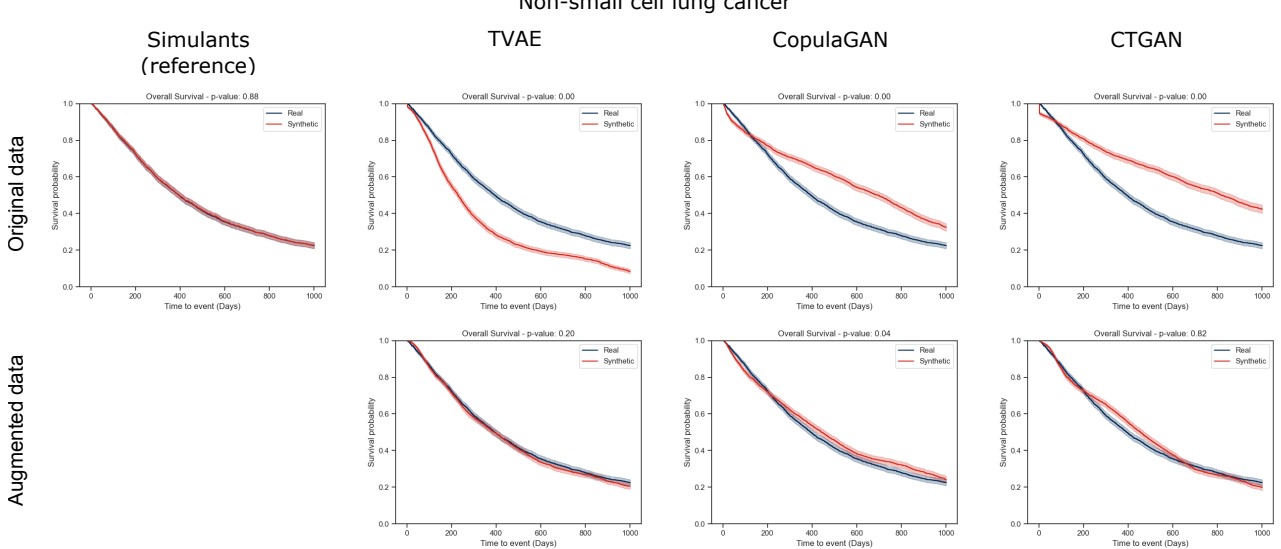

*Figure 2.* Survival probability for Non-small cell lung cancer through synthetic and original clinical trial data. Rows represent results of the Kaplan-Meier curves (Altman, 1992)(MK Goel & Kishore, 2010) of the patients' overall survival evaluating on synthetic dataset generated using (i) original clinical trial data and (ii) augmented clinical trial data (6x). Each column represents the synthetic data generation algorithm used to generate the synthetic data TVAE, CopulaGAN, and CTGAN. Results for Simulants, the data augmentation algorithm, are provided for reference.

*Table 8.* Privacy evaluation for the Multiple Myeloma dataset. Best score across models per algorithm is in **bold**. Best overall score is underlined.

| MODEL | SINGLING OUT | LINKABILITY RISK | INFERENCE RISK | PRIVACY LOSS |
|---|---|---|---|---|
| CTGAN (ORIGINAL) | **0.000** | **0.000** | 0.055 | **0.003** |
| CTGAN (AUGMENTED 2X) | 0.046 | **0.000** | 0.028 | 0.015 |
| CTGAN (AUGMENTED 3X) | 0.056 | 0.143 | **0.011** | -0.003 |
| CTGAN (AUGMENTED 6X) | 0.033 | 0.167 | 0.023 | 0.005 |
| CTGAN (AUGMENTED 11X) | 0.022 | **0.000** | 0.054 | 0.020 |
| COPULAGAN (ORIGINAL) | 0.024 | 0.056 | 0.033 | **0.000** |
| COPULAGAN (AUGMENTED 2X) | **0.000** | **0.000** | 0.056 | 0.010 |
| COPULAGAN (AUGMENTED 3X) | 0.055 | **0.000** | **0.022** | -0.015 |
| COPULAGAN (AUGMENTED 6X) | 0.125 | **0.000** | 0.068 | -0.015 |
| COPULAGAN (AUGMENTED 11X) | **0.000** | 0.040 | 0.029 | 0.001 |
| TVAE (ORIGINAL) | 0.022 | 0.006 | 0.033 | 0.015 |
| TVAE (AUGMENTED 2X) | 0.053 | **0.000** | 0.066 | 0.026 |
| TVAE (AUGMENTED 3X) | **0.000** | **0.000** | **0.010** | **0.011** |
| TVAE (AUGMENTED 6X) | **0.000** | **0.000** | 0.032 | 0.015 |
| TVAE (AUGMENTED 11X) | 0.086 | **0.000** | 0.038 | 0.043 |

*Table 9.* Privacy evaluation for the Non-small cell lung cancer data. Best score across models per algorithm is in **bold**. Best overall score is underlined.

| MODEL | SINGLING OUT | LINKABILITY RISK | INFERENCE RISK | PRIVACY LOSS |
|---|---|---|---|---|
| CTGAN (ORIGINAL) | 0.038 | 0.120 | 0.047 | -0.003 |
| CTGAN (AUGMENTED 2X) | 0.055 | **0.000** | 0.093 | 0.004 |
| CTGAN (AUGMENTED 3X) | **0.000** | **0.000** | 0.104 | 0.004 |
| CTGAN (AUGMENTED 6X) | 0.083 | **0.000** | **0.035** | **-0.001** |
| CTGAN (AUGMENTED 11X) | **0.000** | 0.148 | 0.078 | 0.008 |
| COPULAGAN (ORIGINAL) | 0.063 | **0.000** | **0.041** | 0.004 |
| COPULAGAN (AUGMENTED 2X) | 0.074 | **0.000** | 0.056 | -0.005 |
| COPULAGAN (AUGMENTED 3X) | 0.080 | 0.169 | 0.058 | **-0.001** |
| COPULAGAN (AUGMENTED 6X) | 0.129 | 0.029 | 0.056 | 0.005 |
| COPULAGAN (AUGMENTED 11X) | **0.038** | **0.000** | 0.054 | 0.009 |
| TVAE (ORIGINAL) | 0.061 | 0.063 | 0.092 | 0.009 |
| TVAE (AUGMENTED 2X) | **0.000** | 0.056 | 0.072 | 0.019 |
| TVAE (AUGMENTED 3X) | **0.000** | **0.018** | **0.056** | 0.013 |
| TVAE (AUGMENTED 6X) | **0.000** | 0.141 | 0.071 | 0.006 |
| TVAE (AUGMENTED 11X) | **0.000** | 0.095 | 0.072 | **0.005** |

the generated data can be regarded as pseudo-real training data for the generative models. On the other hand, if the augmentation method does not produce high quality data, the generative model will learn and produce low quality data with distributions that are very different from the source data. Future work will include measuring the relationship and the bounds on performance improvement in relation to the augmentation scaling factor, initial data size (varying from extremely small tens to $10^5$) and other augmentation techniques while also including the privacy-preserving generative algorithms (including PATE-GAN(James Jordon & van der Schaar, 2019), DP-GAN(Liyang Xie & Zhou, 2018)), with a focus on identifying the limitations of this augmentation framework through empirical exploration. For continued development and innovation in clinical trial re-

search, collaboration, while preserving privacy of patients, is essential. This framework provides an efficient way to generate high quality synthetic data at scale while simultaneously preserving the fidelity and privacy interests, increasing the democratization of trial data to support further research into more efficient, higher quality clinical trials.

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

*Table 10.* Privacy evaluation for the CAR-T dataset. Best score across models per algorithm is in **bold**. Best overall score is underlined.

| MODEL | SINGLING OUT | LINKABILITY RISK | INFERENCE RISK | PRIVACY LOSS |
|---|---|---|---|---|
| CTGAN (ORIGINAL) | **0.000** | 0.083 | 0.027 | -0.001 |
| CTGAN (AUGMENTED 2X) | **0.000** | **0.000** | **0.008** | **0.000** |
| CTGAN (AUGMENTED 3X) | 0.062 | **0.000** | 0.025 | 0.003 |
| CTGAN (AUGMENTED 6X) | **0.000** | **0.000** | 0.028 | 0.001 |
| CTGAN (AUGMENTED 11X) | **0.000** | **0.000** | 0.056 | 0.003 |
| COPULAGAN (ORIGINAL) | 0.021 | 0.042 | **0.031** | **-0.001** |
| COPULAGAN (AUGMENTED 2X) | **0.000** | **0.000** | 0.057 | -0.006 |
| COPULAGAN (AUGMENTED 3X) | **0.000** | **0.000** | 0.034 | -0.002 |
| COPULAGAN (AUGMENTED 6X) | **0.000** | **0.000** | 0.041 | 0.002 |
| COPULAGAN (AUGMENTED 11X) | 0.068 | 0.057 | 0.038 | 0.004 |
| TVAE (ORIGINAL) | 0.040 | 0.113 | 0.054 | -0.002 |
| TVAE (AUGMENTED 2X) | 0.011 | **0.000** | 0.059 | -0.003 |
| TVAE (AUGMENTED 3X) | 0.035 | **0.000** | 0.077 | -0.009 |
| TVAE (AUGMENTED 6X) | **0.000** | **0.000** | **0.018** | **-0.001** |
| TVAE (AUGMENTED 11X) | 0.012 | 0.265 | 0.072 | 0.006 |

Beigi, M., Shafquat, A., Mezey, J., and Aptekar, J. W. Synthetic clinical trial data while preserving subject-level privacy. In *NeurIPS 2022 Workshop on Synthetic Data for Empowering ML Research*, 2022.

Bland, J. and Altman, D. The logrank test. *BMJ*, 328:1073, 2004.

Buitinck, L., Louppe, G., Blondel, M., Pedregosa, F., Mueller, A., Grisel, O., Niculae, V., Prettenhofer, P., Gramfort, A., Grobler, J., Layton, R., VanderPlas, J., Joly, A., Holt, B., and Varoquaux, G. API design for machine learning software: experiences from the scikit-learn project. In *ECML PKDD Workshop: Languages for Data Mining and Machine Learning*, pp. 108–122, 2013.

Chi-Heng Lin, Chiraag Kaushik, E. L. D. and Muthukumar, V. The good, the bad and the ugly sides of data augmentation: An implicit spectral regularization perspective. 2022. URL https://arxiv.org/abs/2210.05021.

Cover, T. M. and Hart, P. E. Nearest neighbor pattern classification. *IEEE Transactions on Information Theory*, 13, 1967.

der Maaten L, V. and G., H. Visualizing data using t-sne. *Journal of Machine Learning Research*, 9:2579–2605, 2008. URL https://openreview.net/pdf?id=TGZ8P21_jHL.

Emam, K. E., Mosquera, L., and Zheng, C. Optimizing the synthesis of clinical trial data using sequential trees. *Journal of the American Medical Informatics Association*, 28(1):3–13, 2021.

Engelmann, J. and Lessmann, S. Conditional wasserstein gan-based oversampling of tabular data for imbalanced learning. *Expert Systems with Applications*, 174, 2020. URL https://doi.org/10.1016/j.eswa.2021.114582.

Freedman, D., Pisani, R., and Purves, R. Statistics (international student edition). *Pisani, R. Purves, 4th edn. WW Norton & Company, New York*, 2007.

F.R.S., K. P. Liii. on lines and planes of closest fit to systems of points in space. *The London, Edinburgh, and Dublin Philosophical Magazine and Journal of Science*, 2(11): 559–572, 1901. doi: 10.1080/14786440109462720.

Ghadeer Ghosheh, J. L. and Zhu, T. A review of generative adversarial networks for electronic health records: applications, evaluation measures and data sources. 2022. URL https://arxiv.org/abs/2203.07018.

Giannis Nikolentzos, Michalis Vazirgiannis, C. X. M. L. and Brandt, E. G. Synthetic electronic health records generated with variational graph autoencoders. *NPJ Digital Medicine*, 6, 2023. URL https://doi.org/10.1038/s41746-023-00822-x.

Giomi, M., Boenisch, F., Wehmeyer, C., and Tasnádi, B. A unified framework for quantifying privacy risk in synthetic data, 2023. URL https://petsymposium.org/popets/2023/popets-2023-0055.php.

Gunjan Ansari, M. G. and Saxena, C. Data augmentation for mental health classification on social media. 2021.

H.-H. Wong, A. Jessup, A. S. A. B. A. B. and Eyraud, J. Examination of clinical trial costs and barriers for drug development final. 2014. URL https://aspe.hhs.gov/system/files/pdf/77166/rpt_erg.pdf.

Hanin, B. and Sellke, M. Approximating continuous functions by relu nets of minimal width. *arXiv preprint arXiv:1710.11278*, 2017.

Hung, S.-K. and Gan, J. Q. Augmentation of small training data using gans for enhancing the performance of image classification. 2021. URL 10.1109/ICPR48806.2021.9412399.

James Jordon, J. Y. and van der Schaar, M. Pate-gan: Generating synthetic data with differential privacy guarantees. 2019. URL https://openreview.net/forum?id=S1zk9iRqF7.

Katherine Lee, Daphne Ippolito, A. N. C. Z. D. E. C. C.-B. and Carlini, N. Deduplicating training data makes language models better. 2022. URL arXiv:2107.06499.

Khoshgoftaar, C. S. . T. M. A survey on image data augmentation for deep learning. *J Big Data*, 2019. URL https://doi.org/10.1186/s40537-019-0197-0.

Kulynych, J. Hipaa compliance in clinical trials. *Journal of oncology practice*, 4(1):9, 2008.

Langley, P. Crafting papers on machine learning. In Langley, P. (ed.), *Proceedings of the 17th International Conference on Machine Learning (ICML 2000)*, pp. 1207–1216, Stanford, CA, 2000. Morgan Kaufmann.

Leland McInnes, J. H. and Melville, J. Umap: Uniform manifold approximation and projection for dimension r. https://arxiv.org/abs/1802.03426. accessed: 2023-5-01.

Liyang Xie, Kaixiang Lin, S. W. F. W. and Zhou, J. Differentially private generative adversarial network. 2018. URL https://arxiv.org/abs/1802.06739.

Lloyd, S. Least squares quantization in pcm. *IEEE transactions on information theory*, 28(2):129–137, 1982.

Massey Jr, F. J. The kolmogorov-smirnov test for goodness of fit. *Journal of the American statistical Association*, 46 (253):68–78, 1951.

MK Goel, P. K. and Kishore, J. Understanding survival analysis: Kaplan-meier estimate. *Int J Ayurveda Res.*, 1: 274–278, 2010.

Muhammad Sakib Khan Inan, S. H. and Uddin, M. N. Data augmentation guided breast cancer diagnosis and prognosis using an integrated deep-generative framework based on breast tumor's morphological information. *Informatics in Medicine Unlocked*, 37, 2023. URL https://www.sciencedirect.com/science/article/pii/S2352914823000138.

NIH. "what are clinical trials and studies?". 2023.

Nitesh V. Chawla, Kevin W. Bowyer, L. O. H. and Kegelmeyer, W. P. Smote: synthetic minority oversampling technique. *Journal of Artificial Intelligence Research*, 16:321–357, 2002.

Patki, N., Wedge, R., and Veeramachaneni, K. Sdgym synthesizer benchmark. https://github.com/sdv-dev/SDGym. accessed: 2023-5-01.

Patki, N., Wedge, R., and Veeramachaneni, K. The synthetic data vault. In *2016 IEEE International Conference on Data Science and Advanced Analytics (DSAA)*, pp. 399–410. IEEE, 2016.

Rasool Fakoor, Jonas Mueller, N. E. P. C. A. J. S. Fast, accurate, and simple models for tabular data via augmented distillation. *NeurIPS*, 2020. URL https://arxiv.org/abs/2006.14284.

Shafquat, A., Mezey, J., Beigi, M., Sun, J., and Aptekar, J. W. A source data privacy framework for synthetic clinical trial data. In *NeurIPS 2022 Workshop on Synthetic Data for Empowering ML Research*.

Steven Y. Feng, Varun Gangal, J. W. S. C. S. V. T. M. and Hovy, E. A survey of data augmentation approaches for nlp. *Findings*, 2021.

Tomoki Ishikawa, T. Y. and Urushihara, H. An nlp-inspired data augmentation method for adverse event prediction using an imbalanced healthcare dataset. *IEEE Access*, 10, 2022.

Xu, L., Skoularidou, M., Cuesta-Infante, A., and Veeramachaneni, K. Modeling tabular data using conditional gan. *Advances in Neural Information Processing Systems*, 32, 2019.

Yale, A., Dash, S., Dutta, R., Guyon, I., Pavao, A., and Bennett, K. P. Privacy preserving synthetic health data. In *ESANN 2019-European Symposium on Artificial Neural Networks, Computational Intelligence and Machine Learning*, 2019.

Yuki Eizuka, K. H. and Suzuki, I. Impact of duplicating small training data on gans. 2021.

Z. Azizi, C. Zheng, L. M. L. P. K. E. E. and Collaborators, G.-F. Can synthetic data be a proxy for real clinical trial data? a validation study. *BMJ Open*, 11:e043497, 2021.

