# OpenReview forum: "An interpretable data augmentation framework for improving generative modeling of synthetic clinical trial data"
_ICML.cc/2023/Workshop/IMLH — IMLH 2023 Oral_

### Official Review · Reviewer_6SQK · 2023-06-16
**Good Clarity**

**Rating:** 6
**Confidence:** 3

**Review:**

Pros:
This paper present an interpretable data augmentation framework for improving generative models used to produce synthetic clinical trial data -> apply to three clinical trial datasets -> results show that this framework can improve the quality of synthetic data produced using
generative algorithms

Cons:
1. The font size of the text within figure 2 is too small, making it challenging to read -> a larger font would improve its legibility
2. Table captions could benefit from including brief summaries -> a quick understanding of the table contents
3. there are issues with inconsistent or incorrect capitalization throughout the paper

---

### Official Review · Reviewer_qmro · 2023-06-17
**This paper use data augmentation (Simulants) to improve the generative model.**

**Rating:** 6
**Confidence:** 3

**Review:**

Strengths of the paper:
1) The paper is well-organized and easy to understand.
2) The experiment is clear that the data augmentation method (Simulants) can improve the performance of generative models.

Weakness:
1) From my point of view, the key contribution of this paper is the data augmentation framework, which is to train a generative model using original and augmented data. Is this paper the first time thought about the framework? If it is not, please cite the original reference.
2) It would be nicer if the author can compare other data augmentation methods (except Simulants) in the experiments.
3) Please increase the dpi of Figure 1 and 2.

---

### Official Review · Reviewer_fhQv · 2023-06-17
**Thorough evaluation, should be accepted at workshop.**

**Rating:** 8
**Confidence:** 5

**Review:**

The authors propose to combine "data augmentation" with DL-based tabular data generation models to yield more synthetic realistic clinical trail data. I think the evaluation is quite thorough (many models, oversampling ratios and metrics) and will be of interest to the small yet growing field of tabular data generation.

Thus, presenting this work at this workshop will benefit the community and thus it should be accepted.

There are some key weaknesses that the authors might want to address in future iterations:
- The simulants method by itself scores very highly, or even wins, on all the metrics. The authors should justify better why then using the DL-based generators is necessary at all
- The metrics used as good, standard in the field but many works use only a subset, so this is commendable. However, in my opinion, none of the metrics capture what we _really_ want in realistic synthetic data, especially for the "fidelity" and "utility" metrics. This relates to the point above: the simulants method excels at these metrics, but I don't think it does all one would want out of a synthetic tabular data generation method.

---

### Meta-Review · Program_Chairs · 2023-06-19

**Recommendation:** Accept (Oral)
**Confidence:** 4

**Metareview:**

This paper is relevant to the workshop and contains several merits. The authors can further highlight the work's originality and clarify the other related work in the final version.

---

### Decision · Program_Chairs · 2023-06-20

Accept (Oral)